# Comparative Pathology of West Nile Virus in Humans and Non-Human Animals

**DOI:** 10.3390/pathogens9010048

**Published:** 2020-01-07

**Authors:** Alex D. Byas, Gregory D. Ebel

**Affiliations:** Department of Microbiology, Immunology, and Pathology, College of Veterinary Medicine and Biomedical Sciences, Colorado State University, Fort Collins, CO 80523, USA; alex.byas@colostate.edu

**Keywords:** West Nile virus, pathology, animal models, flavivirus, review

## Abstract

West Nile virus (WNV) continues to be a major cause of human arboviral neuroinvasive disease. Susceptible non-human vertebrates are particularly diverse, ranging from commonly affected birds and horses to less commonly affected species such as alligators. This review summarizes the pathology caused by West Nile virus during natural infections of humans and non-human animals. While the most well-known findings in human infection involve the central nervous system, WNV can also cause significant lesions in the heart, kidneys and eyes. Time has also revealed chronic neurologic sequelae related to prior human WNV infection. Similarly, neurologic disease is a prominent manifestation of WNV infection in most non-human non-host animals. However, in some avian species, which serve as the vertebrate host for WNV maintenance in nature, severe systemic disease can occur, with neurologic, cardiac, intestinal and renal injury leading to death. The pathology seen in experimental animal models of West Nile virus infection and knowledge gains on viral pathogenesis derived from these animal models are also briefly discussed. A gap in the current literature exists regarding the relationship between the neurotropic nature of WNV in vertebrates, virus propagation and transmission in nature. This and other knowledge gaps, and future directions for research into WNV pathology, are addressed.

## 1. Introduction

Twenty years after its introduction, West Nile virus (WNV, *Flaviviridae*: *Flavivirus*) continues to be the leading cause of arboviral disease in the continental United States (US) [1]. WNV was first isolated in 1937 from the blood of a woman in the West Nile district of Uganda [2]. During the 1950s, the virus was isolated from birds, mosquitoes and people in Egypt [3,4]. The virus continued to present primarily as recurrent epidemics of mild febrile disease in Africa, the Middle East and Europe [5]. However, the disease phenotype was drastically different during outbreaks in the United States, Romania, Russia and Israel, which occurred in the late 1990s and early 2000s [6,7]. In the United States, about 1 in 150 clinical patients developed neuroinvasive disease, characterized by meningitis, encephalitis, and/or poliomyelitis [8,9]. Of those with neuroinvasive disease, 9% experienced mortality [10]. The US outbreak represented the first known introduction of WNV to the Western hemisphere. Outbreaks during the late 1990s in Israel and eastern Europe were also characterized by higher rates of fatal neuroinvasive disease [11,12]. Given that these outbreaks occurred in disparate locations, phylogenetic analysis was performed to assess the degree of relatedness between the virus isolates. The strain introduced into the United States (NY99) had greater than 99.8% nucleotide homology to virus isolated from the brain of a goose in Israel from 1998 and human Israeli cases in 1999 [13,14]. Isolates from the US also held similar relatedness to isolates from Romania [13,15]. It is surmised that these closely related viruses were able to cause disease in these distant locations as a result of globalization. Following its arrival in the United States, the virus showed remarkable adaptability to a new environment, quickly spread and is now endemic.

## 2. The Virus and Its Ecology

WNV is a member of the genus *Flavivirus* within the family *Flaviviridae*. The WNV genome is a positive-sense, single-stranded RNA molecule of approximately 11,000 nucleotides. Within host cells, viral RNA is translated and processed into 10 proteins: three structural (envelope, membrane and nucleocapsid) and seven nonstructural (NS) (NS1, NS2A, NS2B, NS3, NS4A, NS4B, and NS5) [13]. The virus has two commonly accepted major genetic lineages and at least five additional proposed lineages [6,12,16,17,18,19,20,21]. Lineage 1 is widely distributed and contains isolates from Africa, Asia, Australia, Europe, the Middle East, India and North America [21,22]. Lineage 2 viruses have been isolated in Africa, Madagascar, the Middle East and Europe [23,24,25,26]. Initially, lineage 1 viruses were considered to be more virulent than lineage 2 viruses; however, it has been shown that both lineages can cause neuroinvasive disease in humans and animals and lineage 2 has now displaced lineage 1 as the primary cause of WNV disease in Europe [23,27,28,29,30,31].

WNV is maintained in nature by ornithophilic *Culex* mosquitoes and wild birds, with the particular mosquito species varying by region in the United States and worldwide (reviewed extensively by Kramer et al. [32]). *Culex tarsalis* drive epidemic transmission in the western United States while *Culex pipiens* is the main vector in the eastern United States. *Culex quinquefasciatus* and *Culex nigripalpus* are important vectors for WNV in the southeastern United States [33,34]. While many avian species have been infected with WNV, house sparrows and American robins have been identified as key to the virus’ maintenance in nature [35,36,37]. When the preferred bird hosts become less abundant during late summer and fall, ornithophilic mosquitoes shift their feeding to mammals leading to human epidemics [38].

In addition to horizontal circulation between birds and mosquitoes, WNV maintenance also has been described in the absence of a mosquito vector. There have been multiple reports of WNV viral RNA detected in carcasses and feces in crow roosts during the winter when mosquitoes are unlikely contributors to transmission, although the transmission mechanism was not specifically elucidated [39,40]. Similar occurrences of WNV disease during the winter have been seen in Bald Eagles, where feeding on the carcasses of infected Eared Grebes was considered the most likely source of infection [41]. Experimentally, non-vector direct transmission also has been demonstrated in geese, with oral and cloacal shedding serving as the most likely sources of viral transmission [42]. Chronic infection may also contribute to winter transmission as infectious virus has been isolated from house sparrows up to 43 days post inoculation [43]. Overwintering of WNV may also occur by vertical transmission from adult female mosquitoes to their progeny, although this process is highly inefficient [44,45]. In addition to the maintenance of WNV in nature via non-vector mechanisms, WNV disease has also been reported in humans potentially as a result of non-vector transmission including percutaneous exposure, transplacental transmission, breast milk consumption, blood transfusion and organ transplant [46,47,48,49,50,51].

Interestingly, despite the presence of numerous bird species and mosquitoes which should allow for its maintenance, WNV is not frequently seen in central and South America. There have been no major outbreaks and reports are rare in humans and horses [52,53,54,55]. Possible causes may include cross-protection due to other circulating flaviviruses or a dilution effect on WNV due to high disease host diversity [56,57]. 

Much of the research surrounding West Nile virus focuses on (a) viral and ecological factors which affect viral transmission and (b) viral and immune factors which determine pathology and disease. A distinction must be made between the ability for transmission and pathology. While numerous species can be infected and experience pathology and disease, birds are well-established as the primary species which develop the high viral titers required to infect mosquitoes and contribute to virus perpetuation.

Experimental animal models have been used to examine the factors that affect WNV transmission. Some of that work has used wild birds, allowing for direct investigation of the most relevant virus ecology [58,59,60]. Reports also demonstrate that additional species including Eastern cottontail rabbits, fox squirrels and alligators have the potential to infect mosquitoes [61,62,63], indicating that continued investigation into possible non-avian contributions to WNV maintenance is warranted. While WNV is primarily transmitted and maintained between birds and mosquitoes, it can infect and cause pathology and disease in a wide range of vertebrates. This plasticity is relatively unique amongst arboviruses and has been demonstrated in both natural and experimental infections. When it comes to the viral and immune factors leading to pathology, much has been examined in mouse models and is briefly described below. The pathology seen in natural and experimental infections will be discussed in detail in this review.

## 3. Natural WNV Disease

### 3.1. Humans

While approximately 80% of people infected with WNV are asymptomatic, the majority of symptomatic patients experience a mild febrile disease lasting up to a week, a syndrome known as West Nile fever (WNF) [64]. In addition to fever, mild disease is characterized by headache, gastrointestinal problems, rash, myalgia, arthralgia and malaise. West Nile neuroinvasive disease (WNND) occurs in less than 1% of infected people and is manifested as multiple syndromes including West Nile meningitis (WNM), West Nile encephalitis (WNE) and West Nile poliomyelitis (WNP) [65]. WNM describes inflammation which is primarily restricted to the meninges, the connective tissue coverings of the brain and spinal cord. WNM is typically associated with more favorable outcomes and is the most common manifestation of neuroinvasive disease in younger patients [66]. WNE is a more invasive disease process, in which the brain parenchyma is infected and inflamed. This is more commonly seen in older adults and the immunosuppressed [10]. Clinical signs associated with WNE range in severity and can include tremors, cerebellar ataxia, and general Parkinsonism [66]. WNP is associated with infection of the anterior horn cells (lower motor neurons) of the spinal cord, resulting in a polio-like flaccid paralysis, which at its most severe can cause quadriplegia and respiratory impairment [67,68,69]. WNP is distinct from a rarer Guillain-Barre syndrome which has been reported in association with WNV infection [65]. While these classifications are important, clinical presentation of WNV infection may present as any mixture of these syndromes. 

Histological findings in West Nile neuroinvasive disease are nonspecific and typical of many viral encephalitides, and these are characterized by perivascular lymphocytic infiltrates, microglial nodules, neuronal loss, and neuronophagia [67,70,71,72]. In severe cases, necrosis can be seen [73]. Most commonly affected regions of the central nervous system (CNS) have extrapyramidal (movement-related) function and include the brainstem (medulla and pons), deep gray matter nuclei (substantia nigra of the basal ganglia and thalamus), and cerebellum with gray matter being the most severely affected [70,73,74,75,76]. In the spinal cord, the anterior horns (ventral horns) and anterior spinal nerve roots are frequently involved and associated lower motor neuron loss results in muscle weakness [71,74,75,77,78]. Clinically observed muscle weakness correlates histologically to neurogenic atrophy of the skeletal muscle [78]. 

While fever and neurological disease are the most well-known clinical manifestations of West Nile virus infection, there are less frequently observed non-neurological clinical findings. Ocular manifestations seen in WNV infection, specifically those associated with the optic nerve and the retina, can be considered an extension of the brain. In one report, 80% of patients with WNV neurologic disease have chorioretinal involvement, primarily multifocal chorioretinitis, although it is often asymptomatic and self-limiting [79,80]. Additional WNV ocular manifestations vitritis, optic neuritis and retinal hemorrhage [81,82,83]. Renal failure has been documented in one study as present in 9% of hospitalized WNV patients and in another study, 21% of deceased WNV patients had renal failure [84,85]. More commonly seen in other vertebrate species, myocarditis has been reported occasionally in humans [86,87,88]. Other rare lesions include hepatitis, pancreatitis, orchitis and myositis [5,89,90,91]. 

There are also sequelae to acute disease which have been observed in both convalescent and fully recovered patients. Subsequent to WNF, WNM and WNE, there are estimates that 50% of patients are affected by neurocognitive or functional impairment [92,93]. Neuropsychological sequelae include memory problems, headache, cognitive dysfunction, depression and fatigue [93,94,95]. Motor skill abnormalities include tremors, fatigue, decreased strength and abnormal reflexes [93,94]. On magnetic resonance imaging (MRI), WNV survivors had significant cortical thinning in the frontal and limbic cortices and regional atrophy in the cerebellum, brain stem, thalamus, putamen and globus pallidus [93]. These neuropsychological and motor skills issues are interpreted to be the result of prolonged or permanent damage to the nervous system. 

There is also some evidence for persistent infection of WNV [96,97,98]. Sequelae from systemic infections have been observed with the kidneys being one of the most common extra-nervous system sites of disease. WNV is able to persist chronically in the kidneys up to seven years and previous infection lead to chronic kidney disease [97,99]. 

### 3.2. Birds

During the introduction of West Nile virus to the United States, mortality observed in the Bronx zoo and surrounding areas showed that avian species across multiple orders could be affected and included common crows, a fish crow, black-billed magpies, a black-crowned night heron, laughing gulls, a mallard duck, Himalayan Impeyan pheasants, a Blyth’s tragopan, Chilean flamingos, guanay cormorants, bronze-winged ducks, a northern bald eagle and a snowy owl [100]. Lesion variability observed between species is likely multifactorial and related to host factors and intrinsic viral factors which depend on virus strain [101]. Specifically, levels of high viremia associated with being an amplifier host have been shown to correlate with mortality in some birds [102]. Interestingly, high levels of mortality do not always correlate with histopathologic signs and this may be a measure of acute infection resulting in death and occurring in such rapid fashion that lesions do not develop histologically [101].

In contrast to humans and other affected species, most major organ systems have been shown to be affected in natural avian WNV infections. Neurological manifestations are indicative of viral encephalitis and are similar to findings seen in humans and other non-host vertebrate species. This encephalitis is characterized as a lymphoplasmacytic meningoencephalitis with an occasional heterophilic component, heterophils being the avian functional equivalent to the neutrophil. Histologically, there is perivascular cuffing, glial nodules and gliosis, neuronal necrosis and occasional hemorrhage. Frequently affected regions of the nervous system include the brain stem and gray matter of the spinal cord, cerebellum and thalamus. In addition to the nervous system, myocarditis is a common lesion in birds [103,104,105]. Inflammation and necrosis have also been reported in the gastrointestinal tract, kidney, spleen, liver, pancreas, lung, adrenal glands, thyroid, thymus, bursa, bone marrow and skeletal muscle [101]. Lesions in naturally infected birds have been previously reviewed in detail by Gamino and Höfle [95]. This chart has been modified to include the most recent literature as well as pathologic findings in birds experimentally infected with WNV (Table 1) [106,107,108,109,110,111,112,113,114,115,116]. Ocular lesions are seen in raptors (red-tailed hawks, Cooper′s hawks, bald eagles, golden eagles, goshawks) and owls (great horned owls, barred owls) and range from lymphoplasmacytic pectenitis and chorioretinal inflammation and scarring to generalized endophthalmitis [110,116,117,118,119]. Vasculitis has occasionally been noted within multiple organs in a variety of affected avian species [103,120,121]. While systemic infection is a hallmark of WNV infection in many species, adult domestic chickens and turkeys do not frequently experience significant disease [122,123]. Age and a developed immune system likely contribute to this refractory nature of adult chickens as young chicks are susceptible and histologic lesions include myocardial necrosis, necrosis, nephritis and pneumonitis, and rare encephalitis [123]. 

The severe systemic infection and mortality observed in avian species have the potential to significantly alter bird populations. Recent literature has demonstrated massive losses in avian biodiversity in North America, with estimates of 29% loss of abundance since 1970 [124]. WNV likely contributes to some avian losses as it has been estimated to negatively affect populations of between one–fifth and one–half of examined North American avian species, with some (but not all) never recovering to pre-WNV levels [125,126,127,128]. WNV’s negative impacts are potentially exacerbated by climate change and anthropogenic factors such as human land use [129,130,131]. 

### 3.3. Horses

Clinical signs of WNV infection in horses, aside from fever, are primarily related to nervous system infection and inflammation. Approximately 20% of infected horses develop clinical neurological signs [132]. Mortality in unvaccinated horses is between 30 and 50%, inclusive of both natural death and elective euthanasia [55,133]. Of equine survivors, remnant neurological signs are present in between 10–20% of horses [134]. The most severe clinical signs in horses include limb ataxia, tetraparesis, paraparesis, recumbency, seizures and death [133,135]. Additional signs include cranial nerve deficits, muscle fasciculations, hyperexcitability and behavioral changes [52,133,136,137]. 

In horses, histologically affected spinal cords are similar to those of humans with polioencephalomyelitis and are characterized by lymphocytes and fewer numbers of macrophages and neutrophils cuffing vessels, glial nodules, and occasional neuronophagia [138]. Ventral and lateral horns of the spinal cord gray matter are most affected [136,138]. In addition to the spinal cord, the gray matter of the midbrain and hindbrain are commonly affected [136]. Perivascular hemorrhage is also seen in horses. The cerebral cortex seems to be least affected. Extraneural disease in horses includes sporadic renal hemorrhage, lymphoid atrophy and myocarditis [138]. 

Several commonalities unite the clinical presentation of WNV in the most affected vertebrates: birds, horses and humans. These include neurotropism characterized by primarily mononuclear inflammation, neuronal necrosis and gliosis which frequently affects gray matter and varies according to host and virus strain. In addition, renal and ocular tropism seems to be a conserved aspect of clinical disease in birds and humans.

### 3.4. Additional Affected Vertebrate Species

In addition to birds, humans and horses, WNV infects and causes disease in an extraordinary array of vertebrate species (see Figure 1). In many of these, clinical disease is solely neurological and thus similar to what is seen in humans and horses. This is observed in numerous single-animal case reports including those of an alpaca, harbor seal, reindeer, Barbary macaque, white-tailed deer and polar bear [139,140,141,142,143,144]. Histologically, these animals had a nonsuppurative meningoencephalitis which frequently and preferentially affected the gray matter of the brainstem and spinal cord. Similarly, convulsions and ataxia in multiple WNV-infected sheep were seen in association with lymphoplasmacytic meningoencephalitis and myelitis characterized by perivascular cuffing and necrosis [139,145,146]. Histopathology from a sheep which is representative of many species is shown in Figure 2. 

While the expected neurological disease is common in dead-end hosts, some species and individual animals have unique disease presentations. In addition to encephalitis, fox squirrels experience myocarditis, which has been mentioned as a common manifestation in birds. [147]. In alligators infected with WNV, systemic disease is sometimes accompanied by inflammatory nodules in the skin composed of lymphocytes and macrophages [148,149,150]. In a case report of a dog, polioencephalomyelitis and myocarditis were accompanied by vasculitis, pancreatitis and plasmacytic synovitis (inflammation of the articular synovial surface) [151]. In an arctic wolf, renal vasculitis was a significant finding [152]. 

While WNV antigen distribution is frequently described in both natural infection in humans and animals and animal experimental infection, immunohistochemical antigen distribution and staining may indicate sites of replication but do not always correlate to pathology or organ dysfunction; therefore, extensive lists of IHC and PCR results will not be included in this review [76,153,154]. Additionally, IHC does not always align with well-documented sites of virus replication [155]. Antigen can be focal and sparse and is typically visible in only 50% of fatal WNV neuroinvasive disease cases, making this modality relatively insensitive [75]. 

## 4. Biomedical Models of WNV Infection

### 4.1. Mice

A vast body of knowledge regarding the pathogenesis of WNV encephalitis has resulted from experimental infections using animal models. Interestingly, wild-type mice are resistant to WNV infection, but all classical laboratory mice strains are susceptible to infections administered by the intracerebral and intraperitoneal routes [156]. This susceptibility has been mapped to a mutation in the *OAS1b* gene, which is regulated by interferon and contributes to viral RNA degradation [157,158,159]. Histopathology in the laboratory mouse is characterized by neuronal necrosis, lymphohistiocytic perivascular infiltrates, glial nodules, neuronal satellitosis and neuronophagia in the cerebral cortex, cerebellum, brainstem, hippocampus and spinal cord [154,160,161]. Commonly used strains of mice include C57BL/6 mice, mice on a C57BL/6 background and C3H/HeN mice. Disease is not limited to the brain in mice. Thymic atrophy has been described in affected mice, and similarly, the thymus has been affected in ducklings and other birds [154,162]. Gastrointestinal lesions in mice including dilation of the stomach and small intestine and villus blunting (shortening) have been seen in association with degeneration and necrosis of intestinal myenteric ganglia [155,163]. This may serve as a model for gastrointestinal dysmotility seen in humans after flavivirus infection [164]. Gastrointestinal disease is also frequently seen in birds. 

An abundance of information about WNV pathogenesis has been yielded from mouse models of infection. After inoculation, initial viral replication is thought to occur in Langerhans dendritic cells of the skin [165]. These infected Langerhans cells then migrate to draining lymph nodes where virus can be transported to the systemic circulation after return through the thoracic and lymphatic ducts [166,167]. At that point a primary viremia allows for viral dissemination to the visceral organs for additional viral replication [168,169]. Virus must then traverse the blood–brain barrier before replication in nervous tissue to cause encephalitis disease. The lesions of encephalitis and poliomyelitis seen in WNV infection cannot be attributed solely to viral replication or the immune response, but rather are the result of both direct viral injury and immune-mediated pathogenesis. Specific work examines the immunologic mechanisms by which lymphocytes, the primary responding cell type, are recruited into the central nervous system where inflammatory mediators exacerbate viral-induced damage [161,170,171,172,173,174,175].

### 4.2. Hamsters

Golden hamsters are excellent models for WNV encephalitis because they mimic human disease in regards to length of viremia, muscle weakness, gastrointestinal signs, respiratory symptoms and clinical signs including tremors [156,176,177,178]. Hamsters infected with WNV develop neurologic symptoms including tremors and hind limb paralysis associated with progressive pathology in the cerebellum and spinal cord [179]. Histologically, this correlates with lymphoplasmacytic inflammation in those nervous system tissues. The significance of axonal transport as a contributor to entrance of WNV to the CNS has also been demonstrated in hamsters [180]. Hamsters also have persistent renal infection, which mimics the chronic renal infection observed in humans [181]. Overall, hamsters provide good models of disease, but mice are more easily genetically manipulated to assess specific alterations to the immune system. 

### 4.3. Non-Human Primates

In addition to mice, some early experimental work examining WNV encephalitis was performed in non-human primates. Depending on the viral strain, intracerebral inoculation resulted in disease ranging from asymptomatic infection to febrile disease to overt encephalitis [182,183]. Histopathology in acute WNV encephalitis includes severe perivascular and diffuse lymphoplasmacytic inflammation, neuronal degeneration and necrosis and glial nodules in the gray matter. Lesions were most frequent in the cerebellum, brainstem and anterior horns of the spinal cord. Chronic lesions also include loss of Purkinje cells in the cerebellum and spinal motor neurons. In contrast, more recent experimental infections examining antibody response to WNV infection in immunocompetent rhesus macaques and marmosets resulted in low to absent viremia and minimal to absent infectious virus detection in the CNS [184,185]. Persistent viral infection of the CNS, kidney and spleen were seen up to 167 days post infection in some non-human primates [183]. Partially as a result of the associated financial costs and ethical implications, current use of non-human primates in WNV research is infrequent. However, they may be used in continued development of a human vaccine. 

### 4.4. Additional Animal Species Used in WNV Biomedical Research

Multiple other species have been used in WNV biomedical research and contributed to varying aspects of the field. Rats were used in some of the earliest experimental work studying WNV and while older rats were resistant to fatal disease, newborn rats were susceptible [186,187]. Histologically these infections were relatively mild and characterized by meningitis and mild inflammatory infiltrates and perivascular cuffs. The most severe lesions were seen in the hippocampus. In contemporary work, the use of rats is primarily related to toxicology studies which assess for the safety of WNV vaccines [188,189,190]. 

As the species most frequently and significantly affected by WNV infections aside from birds and humans, horses have been used in pathogenesis work and in efficacy and safety studies required for approval of WNV equine vaccines [191,192,193,194,195,196]. There are currently four United States Department of Agriculture-licensed equine WNV vaccines available in the United States [197]. 

New Zealand White rabbits have been demonstrated to be appropriate models for non-lethal WNV infections as only weanling rabbits demonstrate severe lesions and clinical symptoms [198]. Rabbits are also commonly used to generate antibodies against WNV which can be utilized in assays such as immunohistochemistry. Other species which have been examined as animal models for WNV infection include pigs, dog and cats [199,200]. Experimental infection of snakes and bats failed to demonstrate significant viremia and disease [201,202]. Young chicks have occasionally been used as models for avian infection as wild birds can be difficult to work with and require trapping and special care [203,204,205,206,207]. 

Importantly, wild birds have been used for experimental WNV infections, not as a model for mammalian disease, but to assess pathogenesis and factors affecting viral evolution and transmission in the most relevant amplifying hosts [58,60,208,209,210,211,212,213,214,215]. This use is crucial as cell culture and computer modelling are insufficient in their representation of complex living systems.

## 5. Summary and Future Directions

One of the more interesting features of the emergence of WNV is its somewhat unique ecological generalism. Whereas most flaviviruses productively infect a relatively restricted subset of animal species, WNV can infect an extraordinarily wide array of vertebrate taxa. This generalism has produced opportunities to learn about viral pathogenesis across taxa and increase knowledge regarding the extent to which pathogenic mechanisms may be conserved. For example, the most notable feature of WNV disease in humans is neurotropism resulting most frequently in encephalitis and, more uncommonly, meningitis and other neurologic syndromes. This set of clinical presentations is largely conserved in horses, birds and mice. The significance of this broad conservation is that human and avian neurological disease can be modeled with an uncommonly high degree of fidelity using mice, which has facilitated improved understanding of mechanisms of neuroinvasion and subsequent immune-mediated injury.

Other clinical syndromes are also present across broadly divergent vertebrate taxa. Renal tropism is a feature of WNV infection in birds that facilitated early surveillance efforts. Several human studies have demonstrated that renal failure may contribute to poor long-term outcomes among WNV survivors [99]. These studies also documented the presence of WNV in urine sediment from individuals with a history of WNV [216]. In some of these patients, WNV was detected up to nine years after acute infection. Similarly, avian kidneys have been found to persistently harbor WNV after natural and experimental infection. WNV, though typically considered a neurotropic virus, also has significant renal tropism that contributes to pathogenesis in humans, and possibly avian hosts. The degree to which renal infection impacts the health of naturally infected hosts, including humans and birds remains to be fully described.

The detection of persistent infection in the kidneys of people and birds raises another important common feature of WNV pathogenesis that is conserved across several vertebrates and has been previously underappreciated. This is the extent to which WNV can persist within vertebrates despite the induction of a strong antibody and cell-mediated immune response. Monkeys, mice, birds and people have all been demonstrated to develop long-term persistent infections after acute WNV. This has also been noted in a human case of Russian spring–summer encephalitis (caused by tick-borne encephalitis virus) that resulted in progressive neurological disease approximately thirteen years after acute infection [217]. Chronic lesions of prior viral and immune-mediated injury may manifest as the human populations previously affected by WNV continue to age. An important question that remains to be answered is whether persistent, chronic infection by WNV and other flaviviruses, years after acute infection, is an underrecognized aspect of viral pathogenesis. 

## Figures and Tables

**Figure 1 pathogens-09-00048-f001:**
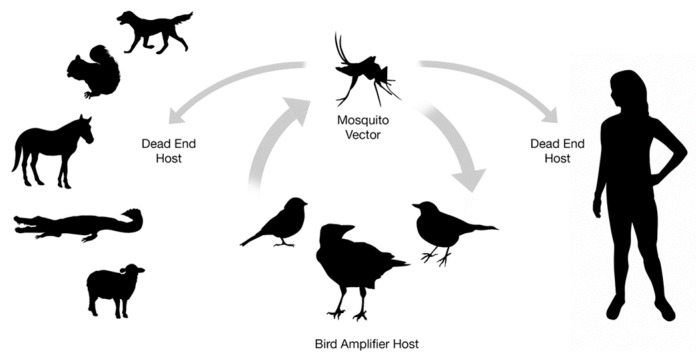
West Nile virus transmission cycle. West Nile virus circulates between *Culex* species mosquitoes and avian amplifying hosts. Humans and a wide array of vertebrate species can be affected as dead-end or non-amplifying hosts.

**Figure 2 pathogens-09-00048-f002:**
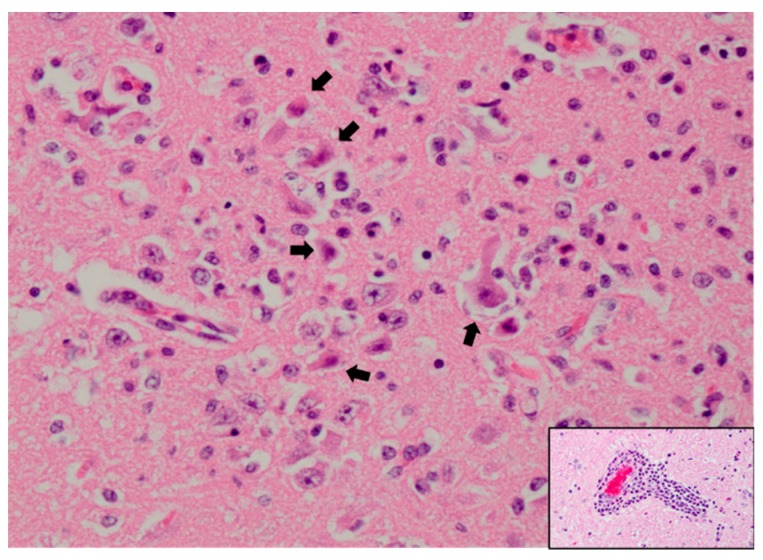
Hippocampus of a WNV-infected sheep with neuronal necrosis and necrotic debris (arrows) and abundant mixed inflammation admixed with remaining neurons. Hematoxylin and eosin staining at 400× magnification. Inset: Perivascular cuffing with lymphocytes and plasma cells. Hematoxylin and eosin staining at 200× magnification. Slide courtesy of Dr. Chad B. Frank, Colorado State University.

**Table 1 pathogens-09-00048-t001:** Distribution of West Nile virus (WNV)-infected bird lesions in natural and experimental infections. Modified from Gamino and Höfle [95].

Order	ACCI	ANSE	CHAR	CICO	FALC	GALL	PASS				PELE	PHOE	PSIT		STRI
Family	ACCI	ANAT	LARI	ARDE	FALC	PHAS	CORV	LANI	PASS	FRIN	PHAL	PHOE	PSIT	STRO	STRI
Brain															
Perivascular cuffs	+	+	+	ND	+	+	+	+	+	−	−	+	+	+	+
Gliosis/glial nodules	+	+	+	ND	+	+	+	+	−	−	−	+	+	+	+
Meningeal inflammation	+	+	−	ND	+	−	+	−	−	−	+	−	−	−	+
Neuronal degeneration and necrosis	+	+	+	ND	+	+	+	−	−	−	+	−	−	+	+
Vasculitis	+	−	−	ND	−	−	+	−	+	−	−	−	−	−	−
Hemorrhage	+	−	−	ND	−	+	−	−	−	−	+	−	−	−	+
Spinal cord															
Inflammation	+	+	ND	ND	ND	ND	ND	NT	ND	NT	ND	ND	ND	NT	+
Neuronal degeneration and necrosis	−	+	ND	ND	ND	ND	ND	NT	ND	NT	ND	ND	ND	NT	+
Peripheral nervous system															
Inflammation	+	−	ND	ND	+	+	−	NT	NT	NT	ND	ND	ND	NT	+
Eye															
Retinitis	+	NT	NT	NT	NT	−	ND	NT	NT	NT	NT	NT	+	NT	+
Retinal necrosis	+	NT	NT	NT	NT	−	ND	NT	NT	NT	NT	NT	−	NT	−
Pectenitis	+	NT	NT	NT	NT	+	ND	NT	NT	NT	NT	NT	−	NT	−
Uveitis (inc. iris, choroid, ciliary body)	+	NT	NT	NT	+	+	ND	NT	NT	NT	NT	NT	−	NT	−
Heart															
Inflammation	+	+	+	+	+	+	+	+	−	−	+	+	+	NT	+
Myofiber necrosis	+	+	−	−	+	+	+	−	−	−	−	−	+	NT	+
Myofibril lysis and mineralization	+	+	+	+	+	+	+	−	−	−	+	+	+	NT	−
Vasculitis	−	−	−	−	+	−	+	−	−	−	−	−	−	NT	−
Hemorrhage	+	−	+	+	−	+	+	−	−	−	+	+	+	NT	−
Gastrointestinal tract															
Inflammation	+	−	ND	ND	−	+	+	−	ND	−	−	−	+	NT	+
Enterocyte necrosis	−	+	ND	ND	−	+	+	−	ND	−	−	−	−	NT	+
Crypt necrosis	−	−	ND	ND	−	−	+	+	ND	−	+	+	+	NT	+
Hemorrhage	−	+	ND	ND	−	−	+	−	ND	−	−	+	−	NT	−
Liver															
Inflammation	+	+	ND	ND	+	+	+	+	−	+	−	−	+	+	+
Hepatocyte necrosis	+	+	ND	ND	−	+	+	+	+	+	+	+	+	+	+
Vasculitis	−	−	ND	ND	+	−	−	−	−	−	−	−	−	−	−
Bile duct hyperplasia	+	−	ND	ND	−	−	−	−	−	−	−	+	−	−	−
Hemosiderosis	+	−	ND	ND	−	+	+	−	+	−	−	−	−	−	+
Hemorrhage	+	−	ND	ND	−	−	+	−	−	−	−	−	−	−	−
Kidney															
Inflammation (interstitial)	+	+	ND	+	+	+	+	ND	+	+	+	+	+	NT	+
Tubular necrosis	+	+	ND	−	−	+	+	ND	+	+	−	−	+	NT	+
Glomerular necrosis	−	−	ND	−	−	−	−	ND	−	−	−	−	+	NT	+
Vasculitis	−	−	ND	−	+	−	−	ND	−	−	−	−	−	NT	−
Hemorrhage	−	+	ND	−	−	−	−	ND	−	−	−	−	−	NT	−
Lung														NT	
Inflammation	+	−	ND	ND	−	+	+	−	+	−	ND	ND	−	NT	+
Necrosis	+	−	ND	ND	−	−	+	−	−	−	ND	ND	−	NT	+
Vasculitis	−	−	ND	ND	−	−	+	−	−	−	ND	ND	−	NT	−
Edema	+	−	ND	ND	+	−	−	+	+	−	ND	ND	−	NT	−
Spleen															
Lymphoid necrosis/apoptosis	+	+	ND	+	−	+	+	+	−	+	+	ND	+	NT	+
Lymphoid depletion	+	+	ND	−	+	+	−	−	−	−	−	ND	−	NT	+
Fibrin deposition	−	−	ND	+	−	+	−	+	−	−	+	ND	−	NT	+
Hemorrhage	−	+	ND	+	−	−	−	−	−	−	+	ND	−	NT	−
Hemosiderosis	+	+	ND	−	−	+	+	−	+	−	−	ND	−	NT	+
Vasculitis	−	−	ND	−	+	−	−	−	−	−	−	ND	−	NT	−
Other lymphoid organs															
Thymic lymphoid necrosis	NT	+	NT	NT	NT	+	NT	NT	NT	NT	NT	NT	−	NT	+
Bursal epithelial atrophy−apoptosis	+	−	NT	NT	NT	ND	NT	NT	−	NT	NT	NT	ND	NT	+
Bursal lymphoid atrophy−apoptosis	+	+	NT	NT	NT	+	NT	NT	+	NT	NT	NT	ND	NT	+
Bone marrow necrosis	ND	−	NT	NT	NT	−	+	NT	NT	NT	NT	NT	ND	NT	ND
Endocrine system															
Pancreatic necrosis	−	+	ND	ND	−	+	−	+	ND	−	−	+	+	NT	+
Pancreatic inflammation	+	+	ND	ND	+	+	−	−	ND	−	+	+	+	NT	+
Adrenal gland necrosis	−	ND	ND	ND	NT	+	−	NT	NT	NT	ND	ND	+	NT	−
Adrenal gland inflammation	+	ND	ND	ND	NT	+	+	NT	NT	NT	ND	ND	+	NT	+
Thyroid gland necrosis	−	+	NT	NT	NT	NT	NT	NT	NT	NT	NT	NT	ND	NT	ND
Thyroid gland inflammation	+	−	NT	NT	NT	NT	NT	NT	NT	NT	NT	NT	ND	NT	ND
Skin															
Inflammation	ND	+	NT	NT	NT	−	NT	ND	NT	−	NT	NT	+	NT	−
Skeletal muscle															
Myofibril degeneration and necrosis	+	−	NT	NT	+	−	ND	NT	NT	−	NT	NT	+	NT	+
Inflammation	+	−	NT	NT	+	+	ND	NT	NT	−	NT	NT	+	NT	+
Fibrosis	+	−	NT	NT	−	−	ND	NT	NT	−	NT	NT	−	NT	+
Gonads															
Inflammation	−	−	ND	ND	−	+	NT	ND	NT	NT	ND	ND	−	NT	+
Necrosis	−	−	ND	ND	−	−	NT	ND	NT	NT	ND	ND	−	NT	+

Order: ACCI: Accipitriformes, ANSE: Anseriformes, CHAR: Charadriiformes, CICO: Ciconiiformes, FALC: Falconiformes, GALL: Galliformes, PASS: Passeriformes, PELE: Pelecaniformes, PHOE: Phoenicopteriformes, PSIT: Psittaciformes, and STRI: Strigiformes. Family: ACCI: Accipitridae, ANAT: Anatidae, LARI: Laridae, ARDE: Ardeidae, FALC: Falconidae, PHAS: Phasianidae, CORV: Corvidae, LANI Laniidae, PASS: Passeridae, FRIN: Fringillidae, PHAL: Phalacrocoracidae, PHOE: Phoenicopteridae, PSIT: Psittacidae, STRO: Strigopidae, and STRI: Strigidae. ND: No described lesion. Tissues were collected at necropsy but no description of lesions (present or absent) is provided. NT: Tissue not tested. Tissue not analyzed in the necropsy. +: Lesion present. Lesion described by at least one author for the tissue. −: Lesion absent. Lesion stated as absent or not specifically described by any author for the tissue.

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
