# Peer review of "Comparative Pathology of West Nile Virus in Humans and Non-Human Animals"

_pathogens, 2020, doi:10.3390/pathogens9010048_

Round 1
Reviewer 1 Report
This is a very comprehensive review article regarding the pathology of WNV in human and other animals.
The authors describe very detail with regard to the experimental animal models of West Nile virus infection.
Author Response
This is a very comprehensive review article regarding the pathology of WNV in human and other animals.
The authors describe very detail with regard to the experimental animal models of West Nile virus infection.
Author Response: No responses necessary. The author appreciates the positive feedback from Reviewer 1
Reviewer 2 Report
The manuscript represents a comprehensive review on comparative pathology of WNV in humans and animals.
The abstract, particularly by its first sentence, leaves an impression that the data in the review refer exclusively or at least mostly to the US although the review also presents data from WNV infections elsewhere. Beside the high number of human cases in the US each year, in Europe, for example, in the 2018 transmission season the total number of reported autochthonous human infections exceeded two thousands. Therefore, in the abstract either all/major heavily affected geographical regions or none of them should be specified when emphasizing the importance of human WNV infection.
In the rows 43 and 44 it should be specified whether the 99.8% homology refers to nucleotide or amino acid homology.
In the description of the virus and its ecology (rows 50-59), specifically the WNV genetic lineages description, more recent references should be used. There are few recent articles comprehensively dealing with this topic.
In the row 190 the term “adult poultry” should be specified using “adult chickens” or similar. The used reference [117] deals specifically with chickens while the term poultry comprises turkeys, ducks, geese etc. which are likely to react differently to WNV from chickens.
In the row 246 “can affected” should be changed into “can be affected” or similar.
In the row 242 the year of publication should be in bold letters.
In the reference section of this manuscript, the Latin names of animals are written in regular font, although in the original titles they are written in italic font.
